

# Muscle activation varies between high-bar and low-bar back squat

Michal Murawa[1], Anna Fryzowicz[1], Jaroslaw Kabacinski[1], Jakub Jurga[1], Joanna Gorwa[1], Manuela Galli[2] and Matteo Zago[2]

[1] Department of Biomechanics, Poznan University of Physical Education, Poznan, Poland
[2] Dipartimento di Elettronica, Informazione e Bioignegneria, Politecnico di Milano, Milano, Italy

## ABSTRACT

**Background.** Differences in the muscular activity between the high-bar back squat (HBBS) and the low-bar back squat (LBBS) on the same representative group of experienced powerlifters are still scarcely investigated. The main purpose of the study was to compare the normalized bioelectrical activity and maximal angles within single homogeneous group between the HBBS and LBBS for 60% one repetition maximum (1RM), 65% 1RM and 70% 1RM.

**Methods.** Twelve healthy men (age 24.3 $\pm$ 2.8 years, height 178.8 $\pm$ 5.6 cm, body mass 88.3 $\pm$ 11.5 kg), experienced in powerlifting performed HBBS and LBBS with comparable external loads equal 60% 1RM, 65% 1RM, and 70% 1RM. Electromyography (EMG) signals of muscle groups were synchronously recorded alongside kinematic data (joints angle) by means of a motion capture system.

**Results.** EMG activity during eccentric phase of squat motion were significantly higher during LBBS than in HBBS for all selected muscles (60% 1RM and 65% 1RM) ($p < 0.05$). All examined muscles were more activated during concentric phase of the squat cycle ($p < 0.05$). In the concentric phase, significant differences between the loads were generally not observed between just 5% 1RM change in load level for LBBS.

**Conclusions.** Our results confirmed significant differences in muscles activation between both squat techniques. Muscle activity during eccentric phase of squat motion were significantly higher during LBBS than HBBS. The differences are crucial for posterior muscle chain during eccentric phase of squat cycle.

# INTRODUCTION

The squat is considered one of the most common strength and conditioning exercises in numerous sport disciplines for both professional and amateur athletes. The main reason for that is the existence of strong association between the squat maximum repetition and increased performance in various athletic tasks. The squat is used primarily to improve muscle strength and power performance of the hip and knee extensors and it is even more effective when performed with external load such as barbell (*Wisloff et al., 2004*; *Usui et al., 2016*; *Wirth et al., 2016*). Due to its multi-joint characteristics, it is also recognized as a screening test for movement deficits (*Kritz, Cronin & Hume, 2009*; *Myer et al., 2014*; *Kushner et al., 2015*; *Rabin & Kozol, 2017*) or even physical examination (*Ahankoob et al.,*

Corresponding author
Michal Murawa,
murawa@awf.poznan.pl

*2011*). Different aspects of squat technique, including muscle activity, were subjects of many research projects and biomechanical analysis (*Gullett et al., 2009*; *Schwanbeck, Chilibeck & Binsted, 2009*; *McBride et al., 2010*; *Fujita et al., 2011*; *Bryanton et al., 2012*; *Clark, Lambert & Hunter, 2012*; *Contreras et al., 2015*; *Contreras et al., 2016*; *Saeterbakken, Andersen & vanden Tillaar, 2016*; *Hammond et al., 2019*). Among the groups which are most interested in the practical application of such results are powerlifters and their coaches. This is due to their competition goal which is lifting one repetition maximum (1RM) weight, so any tip such as training or lifting technique modifications that can improve performance can become crucial.

Powerlifters perform two main techniques of back squat with weights during their trainings: the high-bar back squat (HBBS) and the low-bar back squat (LBBS) (*Wretenberg, Feng & Arborelius, 1996*; *Glassbrook et al., 2019*). The names of the variations describe the barbell position which is held either at the top of the trapezius muscle (just below the process of the C7 vertebra) (HBBS) or further down on the back along the spine of the scapula and over the posterior deltoid (LBBS) (*Wretenberg, Feng & Arborelius, 1996*; *Glassbrook et al., 2017*). Even if LBBS usually allows to lift heavier loads, the HBBS still remains one of the most important exercise in athletes training. The LBBS is characterized by more forward torso position, decreased moment arm due to placing the bar lower on the back and higher activation of posterior muscles group (*Glassbrook et al., 2017*). The benefits of HBBS are more upright torso position, greater ranges of motion for ankle and knee joints which could result in higher activation of quadriceps muscles (*Glassbrook et al., 2017*). However, it is still not well evidenced what are the differences in muscular activity of lower part of the body between HBBS and LBBS (*Glassbrook et al., 2017*). To date, researchers did not analyze the differences in muscle activity between the HBBS and LBBS on the same representative group of experienced powerlifters.

The main aim of this study was to verify existing differences in electromyography (EMG) signal data for selected muscles involved in HBBS and LBBS performance and provide some strong evidence on muscular activation differences between the two squat variations. The authors expect to confirm that LBBS technique is more efficient than HBBS for posterior muscles of lower extremities, but also want to assess the scale of existing differences in experienced homogeneous group of powerlifters. The additional aim was to evaluate the influence of barbell weight level on selected muscle activity and verify if 5% change of load is enough to notice significant muscle activity differences in LBBS or HBBS.

## METHODS

### Participants

Twelve healthy men (age: $24.3 \pm 2.8$ years, height $178.8 \pm 5.6$ cm, body mass (BM) $88.3 \pm 11.5$ kg, BMI $27.5 \pm 2.7$ kg/m$^2$) were selected to participate in the experiment. Seven of them were competitive powerlifters (National Academic Championships) and five of them were preparing for their first competition. They were classified as experienced in resistance training and performing squats (training experience $5.0 \pm 1.7$ years, $1RM_{HBBS}$/BM $1.6 \pm 0.3$, $1RM_{LBBS}$/BM $1.7 \pm 0.2$) (*Earp et al., 2016*; *Banyard et al., 2017*; *Shariat et al., 2017*;

*Hammer, Linton & Hammer, 2018*). 1RM for HBBS and LBBS were set separately during training sessions (one week apart). All participants did not experience any injury incidents in the previous two years, they refrained from lower body training for 48 h before testing and they were able to squat with maximal effort. This project was approved by the Bioethical Committee of the Poznan University of Medical Sciences (number 546/16) and all subjects gave written informed consent to participate in this study.

## Experimental procedures

The experiment was performed in three sessions (one week apart). The first was planned to test participants HBBS 1RM, the second to test LBBS 1RM and the third to evaluate powerlifters muscle activation during HBBS and LBBS with comparable relative external loads equal to 60%, 65% and 70% of subjects' 1RM. External load level was limited to 70% of 1RM in order not to interfere with individual training preparations to the National Academic Championships (*Issurin, 2010*) and also to allow athletes perform their optimum technique. 1 RM testing was followed after 5 min of general warm-up and stretching exercises. The athletes performed 8 repetitions at approximately 50% of 1RM followed by 3 repetitions at 70% of 1RM and then single repetitions with gradually heavier loads until failure. 1RM testing was consistent with acknowledged guidelines as described by *Niewiadomski et al. (2008)*. All participants performed their squats in standardized powerlifting shoes (adidas powerlift 3).

The third session (lasting approximately two hours) started from preparations for EMG data collecting in accordance with SENIAM recommendations (*Kasman et al., 1998*; *Hermens et al., 2000*). Before electrodes placement, the skin area was cleaned with alcohol and shaved if needed. Pairs of Ag/AgCl electrodes (SORIMEX, Poland, 1 cm diameter), were placed bilaterally in a bipolar configuration along the longitudinal axis of lumbar erector spinae (LES), gluteus maximus (GM), long head of biceps femoris (BF), rectus femoris (RF), vastus lateralis (VLO) and vastus medialis (VMO).

The inter-electrode distance (center to center) was 2 cm. Proper placement was confirmed with manual muscle testing and visual inspection of the raw EMG signal. The ground electrode was placed over the posterior superior iliac spine. A set of 19 reflective markers were then fixed by the same investigator on anatomical landmarks according to Vaughan-Davis model: sacrum between posterior superior iliac spines, anterior superior iliac spines, femoral greater trochanter, femoral lateral epicondyle, the head of fibula, lateral malleolus, calcaneal tuber, the head of the fifth metatarsal and markers on the bar on the lateral side of the thigh and shank (*Davis et al., 1991*; *Vaughan, Davis & O'Connor, 1992*) (Fig. 1).

A short dynamic stretching preceded the set of two series of free body weight squat (FBWS) with hands in front (ten repetitions each). If no complaints were reported due to measuring instrumentations or any other cause, the third series of FBWS (seven repetitions) was recorded and then used to normalize the EMG signal. The mean EMG taken from three middle repetitions of FBWS for each muscle was used as a squat reference value (SRV). The main part of the experiment began with the HBBS technique. Each participant performed a short warm up squat series gradually increasing barbell weight, then proceeded to the
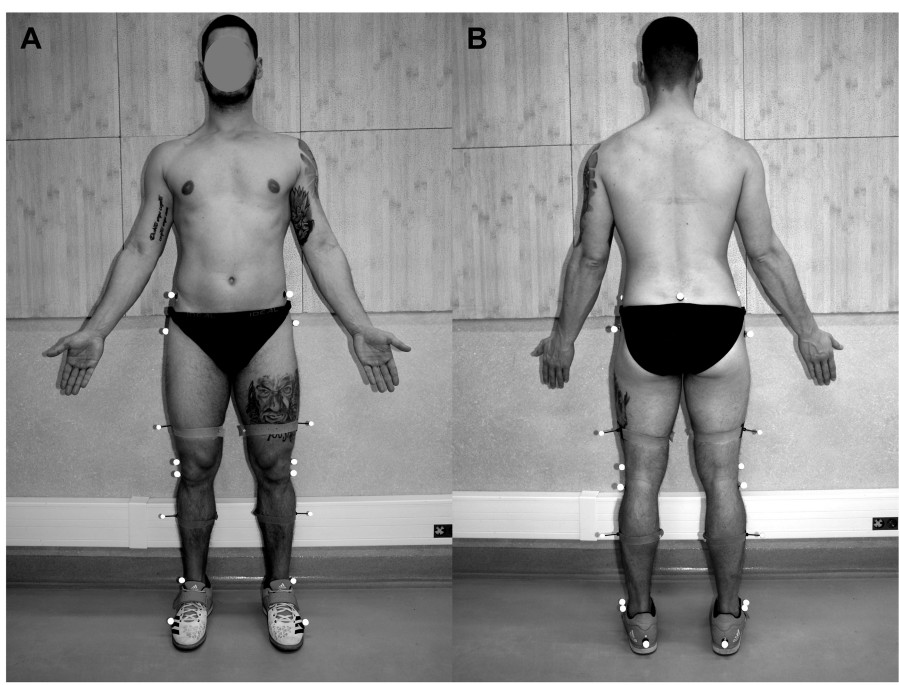

**Figure 1** Placement of reflective markers (A, anterior; B, posterior).

nine HBBS testing trials (3 × 60%, 3 × 65% and 3 × 70% of subject's 1RM). The same procedure was performed for LBBS technique after 30 min rest. All squats were visually inspected to confirm the proper technique and depth (thigh parallel with the floor or lower) (*Aspe & Swinton, 2014*; *Technical Rules Book, 2019*).

The squat cycle (SC), its eccentric and concentric phases, depth, anterior pelvis tilt (PT), hip (HFE), knee (KFE) and ankle (AFE) joint angles (on the sagittal plane) were determined with the use of a motion capture optoelectronic system BTS Smart-D 200 Hz (BTS Bioengineering, Milan, Italy). The eccentric phase of the SC started from the highest vertical position of the marker (set on the sacrum bone between posterior superior iliac spines) and ended in its lowest vertical position. The concentric phase was defined respectively from the lowest position to the highest. The set of FBWS was used for the EMG data normalization. All tests were performed at a self-selected cadence.

## Instrumentation

A Telemyo 2400T G2 device (Noraxon, USA) integrated and synchronized with optoelectronic system BTS Smart-D (BTS Bioengineering, Milan, Italy) was used to record surface EMG activity. The EMG signal was sampled at 800 Hz and then bandpass filtered (bandwidth: 10–400 Hz). EMG signal processing was performed with MyoResearch XP Master Edition software (Noraxon, USA). Artefacts and noise were visually inspected. The EMG signal was full-wave rectified and smoothed using root mean square algorithm (RMS) with 50-ms windows. The peak and mean EMG values during the eccentric and concentric phases of HBBS and LBBS were calculated and presented in %SRV.

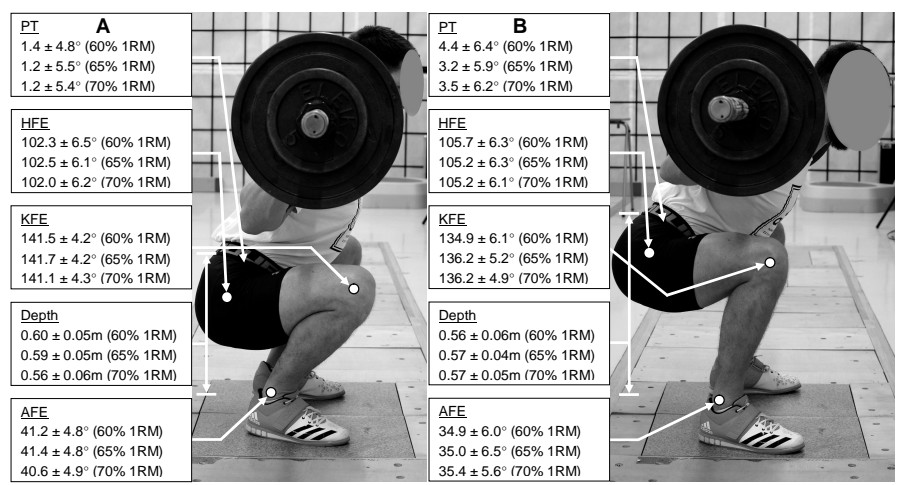

| PT | A |
|---|---|
| 1.4 ± 4.8° (60% 1RM) | |
| 1.2 ± 5.5° (65% 1RM) | |
| 1.2 ± 5.4° (70% 1RM) | |

HFE
102.3 ± 6.5° (60% 1RM)
102.5 ± 6.1° (65% 1RM)
102.0 ± 6.2° (70% 1RM)

KFE
141.5 ± 4.2° (60% 1RM)
141.7 ± 4.2° (65% 1RM)
141.1 ± 4.3° (70% 1RM)

Depth
0.60 ± 0.05m (60% 1RM)
0.59 ± 0.05m (65% 1RM)
0.56 ± 0.06m (70% 1RM)

AFE
41.2 ± 4.8° (60% 1RM)
41.4 ± 4.8° (65% 1RM)
40.6 ± 4.9° (70% 1RM)

PT B
4.4 ± 6.4° (60% 1RM)
3.2 ± 5.9° (65% 1RM)
3.5 ± 6.2° (70% 1RM)

HFE
105.7 ± 6.3° (60% 1RM)
105.2 ± 6.3° (65% 1RM)
105.2 ± 6.1° (70% 1RM)

KFE
134.9 ± 6.1° (60% 1RM)
136.2 ± 5.2° (65% 1RM)
136.2 ± 4.9° (70% 1RM)

Depth
0.56 ± 0.06m (60% 1RM)
0.57 ± 0.04m (65% 1RM)
0.57 ± 0.05m (70% 1RM)

AFE
34.9 ± 6.0° (60% 1RM)
35.0 ± 6.5° (65% 1RM)
35.4 ± 5.6° (70% 1RM)

**Figure 2  Mean ± SD of the PT, HFE, KFE, AFE and depth (A, for HBBS; B, for LBBS).**

Squat depth, PT, HFE, KFE and AFE for the lowest position of the sacrum marker together with the SC temporal characteristics were calculated in Smart Analyzer (BTS Bioengineering, Milan, Italy) using Euler angles convention (Fig. 2) (*Davis et al., 1991*).

### Statistical analyses

Intraclass correlation coefficients (ICCs) for the independent variables from the 3 trials to determine test-retest reliability were calculated (95% confidence interval). ICCs were considered as being *poor* (less than 0.5), *moderate* (between 0.5 and 0.75), *good* (between 0.75 and 0.9) and *excellent* (greater than 0.9) (*Koo & Li, 2016*). Repeated measures analysis of variance (ANOVA) was performed for angles (2 × 2 × 3, technique [HB or LB] × LE [left or right] ×1RM [60%, 65% or 70%]) as well as for bioelectrical activity (2 × 2 × 2 × 3, technique [HB or LB] × LE [left or right] × contraction [eccentric or concentric] ×1RM [60%, 65% or 70%]). A Bonferroni adjustment was used to examine differences between within-subject factors. Sphericity was evaluated using the Mauchly test and Geisser-Greenhouse adjustments were made when sphericity was violated. Effect size estimates for the ANOVA test were determined by the partial eta-squared ($\eta^2$); $\eta^2$ values were interpreted according to the Cohen guidelines of *small* (0.01), *medium* (0.06) and *large* (0.14) (*Cohen, 1988*). For all analyses, the alpha level was set at $p < 0.05$. All statistical analyses were performed in SPSS Statistics software for Windows (version 24.0, IBM Corp, Armonk, NY, USA).

### RESULTS

The mean ± standard deviation (SD) of angles and depth for HBBS and LBBS at the 60% 1RM, 65% 1RM and 70% 1RM are presented in the Fig. 2. Moreover, Fig. 3 illustrates the mean values of the bioelectrical activity for HBBS and LBBS at the 60% 1RM, 65% 1RM and 70% 1RM during the eccentric and concentric phases.
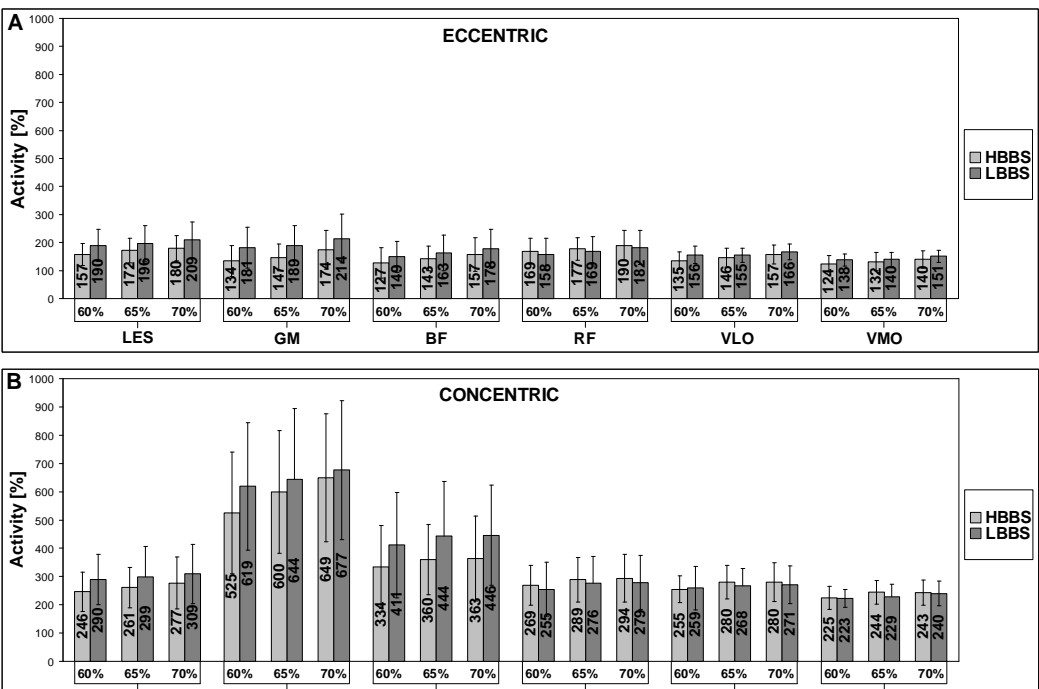

**Figure 3  Bioelectrical activity values of the LES, GM, BF, RF, VLO and VMO for HBBS and LBBS (A, eccentric phase; B, concentric phase).**

## Kinematics

The ICCs values for the mean of 3 trials for the angles ranged from 0.89 to 0.99 ($p < 0.001$) indicated good and excellent reliability. Considering the technique factor, the analysis of the main effect demonstrated significant differences in the values of AFE ($F_{1,30} = 28.19$; $\eta^2 = 3302.56$; $p < 0.001$), KFE ($F_{1,30} = 33.89$; $\eta^2 = 2878.52$; $p < 0.001$), HFE ($F_{1,30} = 56.65$; $\eta^2 = 859.96$; $p < 0.001$) and PT ($F_{1,30} = 24.78$; $\eta^2 = 545.32$; $p < 0.001$). In the case of 1RM factor, was found significant main effect for KFE ($F_{2,60} = 33.3$; $\eta^2 = 18.66$; $p = 0.043$) and HFE ($F_{2,60} = 4.29$; $\eta^2 = 6.50$; $p = 0.018$). In addition, was noted significant interaction effect between the technique and 1RM factors for AFE ($F_{2,60} = 6.60$; $\eta^2 = 16.50$; $p = 0.003$), KFE ($F_{2,60} = 3.78$; $\eta^2 = 20.46$; $p = 0.029$) and HFE ($F_{2,60} = 3.30$; $\eta^2 = 4.85$; $p = 0.044$). Overall, angles displayed a large effect size.

The percentage differences in the angle values between the HBBS and LBBS, and $p$-values of post-hoc test are shown in Table 1. Pairwise comparisons indicated significantly higher values of (1) AFE and KFE for HBBS than LBBS, and (2) PT for LBBS than HBBS ($p < 0.001$). Analysis showed also no significant differences between the 60% 1RM, 65% 1RM and 70% 1RM ($p > 0.05$) as well as between the left and right lower extremities ($p > 0.05$) for all angles.

**Table 1** Angle and activity differences between the HBBS and LBBS for 60% 65% and 70% 1RM (mean ± SD) and *p*-values.

| Variable | 60% 1RM | | 65% 1RM | | 70% 1RM | |
|---|---|---|---|---|---|---|
| | Diff (%) | *p* | Diff (%) | *p* | Diff (%) | *p* |
| **Angle** | | | | | | |
| AFE | 14.3 ± 12.5* | <0.001 | 14.7 ± 11.3* | <0.001 | 12.2 ± 9.0* | <0.001 |
| KFE | 4.6 ± 3.0* | <0.001 | 3.8 ± 2.7* | <0.001 | 3.5 ± 2.7* | <0.001 |
| HFE | −3.2 ± 2.4 | 0.158 | −2.5 ± 1.6 | 0.086 | −3.0 ± 1.6 | 0.106 |
| PT | −30.6 ± 20.6* | <0.001 | −26.4 ± 18.3* | <0.001 | −33.3 ± 12.6* | <0.001 |
| **Activity, eccentric** | | | | | | |
| LES | −15.8 ± 5.1* | <0.001 | −9.5 ± 5.4* | <0.001 | −11.7 ± 5.8* | <0.001 |
| GM | −25.2 ± 8.2* | <0.001 | −20.7 ± 11.1* | <0.001 | −16.8 ± 10.4* | <0.001 |
| BF | −11.8 ± 9.7* | <0.001 | −9.4 ± 6.0* | <0.001 | −10.8 ± 5.4* | <0.001 |
| RF | 8.4 ± 4.9* | 0.001 | 6.2 ± 3.4* | 0.024 | 5.3 ± 4.3* | 0.028 |
| VLO | −13.8 ± 7.7* | <0.001 | −6.8 ± 4.7* | <0.001 | −4.2 ± 10.4 | 0.372 |
| VMO | −11.2 ± 8.0* | <0.001 | −6.7 ± 5.6* | <0.001 | −8.7 ± 5.9* | <0.001 |
| **Activity, concentric** | | | | | | |
| LES | −14.1 ± 6.6* | <0.001 | −9.7 ± 7.3* | <0.001 | −9.9 ± 6.2* | <0.001 |
| GM | −15.8 ± 10.9* | 0.021 | −6.3 ± 4.7* | 0.003 | −4.2 ± 3.0* | 0.005 |
| BF | −17.9 ± 8.5* | <0.001 | −14.1 ± 10.0* | <0.001 | −17.3 ± 7.4* | <0.001 |
| RF | 7.5 ± 3.4* | 0.043 | 6.1 ± 4.7* | 0.020 | 6.3 ± 4.6* | 0.001 |
| VLO | 0.4 ± 5.0 | 0.607 | 4.5 ± 3.4* | <0.001 | 3.5 ± 1.9* | <0.001 |
| VMO | −0.2 ± 4.9 | 0.656 | 6.3 ± 5.0* | <0.001 | 1.0 ± 2.6 | 0.348 |

**Notes.**

HBBS, high-bar back squat; LBBS, low-bar back squat; 1RM, one repetition maximum; SD, standard deviation; Diff, difference; AFE, ankle flexion extension angle; KFE, knee flexion extension angle; HFE, hip flexion extension angle; PT, pelvis tilt angle; LES, lumbar erector spinae; GM, gluteus maximus; BF, biceps femoris; RF, rectus femoris; VLO, vastus lateralis; VMO, vastus medialis.

*significant differences ($p \leq 0.05$) between the HBBS and HBBS.

## EMG

The ICCs values for the mean of 3 trials for the bioelectrical activity ranged from 0.87 to 0.99 ($p < 0.001$) indicated good and excellent reliability. The analysis revealed significant main effect of the technique factor for LES (F $_{1,28}$ = 74.13; $\eta^2$ = 19.77; $p < 0.001$), GM (F$_{1,28}$ = 77.46; $\eta^2$ = 49.49; $p < 0.001$), BF (F$_{1,28}$ = 75.00; $\eta^2$ = 62.11; $p < 0.001$) and VLO (F $_{1,28}$ = 12.42; $\eta^2$ = 0.89; $p = 0.001$). Significant main effect was also observed for 1RM factor for LES (F$_{2,56}$ = 91.80; $\eta^2$ = 4.65; $p < 0.001$), GM (F$_{2,56}$ = 102.73; $\eta^2$ = 34.68; $p < 0.001$), BF (F$_{2,56}$ = 46.00; $\eta^2$ = 7.44; $p < 0.001$), RF (F$_{2,56}$ = 38.90; $\eta^2$ = 4.94; $p < 0.001$), VLO (F$_{2,56}$ = 85.07; $\eta^2$ = 2.59; $p < 0.001$) and VMO (F $_{2,56}$ = 190.96; $\eta^2$ = 1.01; $p < 0.001$). Moreover, significant interaction effect between the technique and 1RM factors for LES was found (F$_{2,56}$ = 4.36; $\eta^2$ = 0.09; $p = 0.017$), GM (F $_{2,56}$ = 23.51; $\eta^2$ = 1.63; $p < 0.001$), VLO (F$_{2,56}$ = 6.11; $\eta^2$ = 0.58; $p = 0.004$) and VMO (F$_{2,56}$ = 10.50; $\eta^2$ = 0.05; $p < 0.001$). Results of the $\eta^2$ for bioelectrical activity showed generally large effect size. The percentage differences in the bioelectrical activity values between the HBBS and LBBS, and *p*-values of post-hoc test are shown in Table 1. Pairwise comparisons demonstrated significantly lower values (1) for HBBS than LBBS for LES, GM and BF (all 1RM; eccentric

**Table 2** Activity differences between the 60%, 65% and 70% 1RM for HBBS and LBBS (mean ± SD) and $p$-values.

| Technique | 60 vs. 65% 1RM | | 65 vs. 70% 1RM | | 60 vs. 70% 1RM | |
|---|---|---|---|---|---|---|
| | Diff (%) | $p$ | Diff (%) | $p$ | Diff (%) | $p$ |
| HBBS, eccentric | | | | | | |
| LES | $-8.4 \pm 4.3^*$ | <0.001 | $-4.3 \pm 3.5^*$ | <0.001 | $-12.4 \pm 4.4^*$ | <0.001 |
| GM | $-10.7 \pm 9.3^*$ | <0.001 | $-14.0 \pm 7.5^*$ | <0.001 | $-23.7 \pm 7.5^*$ | <0.001 |
| BF | $-12.6 \pm 7.5^*$ | <0.001 | $-7.5 \pm 4.1^*$ | 0.012 | $-19.7 \pm 8.3^*$ | <0.001 |
| RF | $-5.5 \pm 4.7^*$ | 0.002 | $-4.9 \pm 3.7^*$ | 0.003 | $-10.1 \pm 6.0^*$ | <0.001 |
| VLO | $-7.4 \pm 5.4^*$ | <0.001 | $-7.1 \pm 5.3^*$ | <0.001 | $-14.1 \pm 4.9^*$ | <0.001 |
| VMO | $-5.9 \pm 4.5^*$ | <0.001 | $-5.8 \pm 4.7^*$ | <0.001 | $-11.5 \pm 3.8^*$ | <0.001 |
| HBBS, concentric | | | | | | |
| LES | $-5.5 \pm 3.3^*$ | 0.001 | $-3.9 \pm 2.4^*$ | 0.050 | $-9.4 \pm 7.0^*$ | <0.001 |
| GM | $-13.2 \pm 5.8^*$ | <0.001 | $-7.4 \pm 4.1^*$ | <0.001 | $-19.8 \pm 8.5^*$ | <0.001 |
| BF | $-10.4 \pm 8.6^*$ | <0.001 | $1.0 \pm 3.3$ | 0.998 | $-9.7 \pm 6.3^*$ | <0.001 |
| RF | $-6.4 \pm 3.1^*$ | <0.001 | $-1.5 \pm 4.4$ | 0.428 | $-7.9 \pm 4.2^*$ | <0.001 |
| VLO | $-8.5 \pm 5.5^*$ | <0.001 | $0.8 \pm 2.4$ | 0.996 | $-7.6 \pm 4.0^*$ | <0.001 |
| VMO | $-7.9 \pm 4.4^*$ | <0.001 | $0.7 \pm 1.8$ | 1.000 | $-7.3 \pm 3.5^*$ | <0.001 |
| LBBS, eccentric | | | | | | |
| LES | $-1.7 \pm 3.1$ | 0.064 | $-6.5 \pm 4.9^*$ | <0.001 | $-8.1 \pm 5.8^*$ | <0.001 |
| GM | $-5.9 \pm 5.1^*$ | 0.021 | $-9.5 \pm 5.3^*$ | <0.001 | $-15.2 \pm 8.4^*$ | <0.001 |
| BF | $-10.4 \pm 6.4^*$ | <0.001 | $-8.8 \pm 5.1^*$ | <0.001 | $-17.9 \pm 12.4^*$ | <0.001 |
| RF | $-7.2 \pm 3.1^*$ | 0.008 | $-6.6 \pm 4.6^*$ | 0.001 | $-13.5 \pm 5.4^*$ | <0.001 |
| VLO | $-0.0 \pm 2.4$ | 1.000 | $-5.2 \pm 4.6^*$ | <0.001 | $-5.0 \pm 5.0^*$ | <0.001 |
| VMO | $-1.2 \pm 1.7$ | 0.132 | $-8.0 \pm 4.9^*$ | <0.001 | $-9.0 \pm 5.3^*$ | <0.001 |
| LBBS, concentric | | | | | | |
| LES | $-1.2 \pm 4.7$ | 0.273 | $-3.7 \pm 3.3$ | 0.072 | $-5.0 \pm 3.8^*$ | 0.001 |
| GM | $-3.3 \pm 3.1$ | 0.061 | $-5.2 \pm 4.3^*$ | 0.003 | $-8.6 \pm 5.8^*$ | <0.001 |
| BF | $-7.0 \pm 5.8^*$ | <0.001 | $-1.9 \pm 4.3$ | 0.997 | $-8.9 \pm 5.7^*$ | <0.001 |
| RF | $-7.7 \pm 3.8^*$ | 0.001 | $-1.5 \pm 2.4$ | 0.647 | $-9.1 \pm 5.8^*$ | <0.001 |
| VLO | $-4.3 \pm 4.0$ | 0.059 | $-0.8 \pm 2.5$ | 0.516 | $-5.1 \pm 3.4^*$ | 0.001 |
| VMO | $-1.5 \pm 3.5$ | 0.267 | $-4.6 \pm 3.7^*$ | <0.001 | $-6.0 \pm 3.7^*$ | <0.001 |

**Notes.**

HBBS, high-bar back squat; LBBS, low-bar back squat; 1RM, one repetition maximum; SD, standard deviation; Diff, difference; LES, lumbar erector spinae; GM, gluteus maximus; BF, biceps femoris; RF, rectus femoris; VLO, vastus lateralis.

*significant differences ($p \le 0.05$) between the 60% 65% and 70% 1RM.

and concentric), VLO (60% 1RM and 65% 1RM; eccentric) and VMO (all 1RM; eccentric), (2) for LBBS than HBBS for RF (all 1RM; eccentric and concentric), VLO (65% 1RM and 70% 1RM; concentric) and VMO (65% 1RM; concentric) ($p < 0.05$). The percentage differences in the bioelectrical activity between the 60% 1RM, 65% 1RM and 70% 1RM, and $p$-values of post-hoc test are presented in Table 2. Comparisons showed significantly lower values (1) at the 60% 1RM than 65% 1RM for all muscles (HBBS; eccentric and concentric), GM (LBBS; eccentric), and BF and RF (LBBS; eccentric and concentric), (2) at the 65% 1RM than 70% 1RM for all muscles (HBBS and LBBS; eccentric), LES (HBBS; concentric), GM (HBBS and LBBS; concentric), and VMO (LBBS; concentric), (3) at the

**Table 3 Activity differences between the eccentric and concentric contractions for HBBS and LBBS (mean ± SD) and p-values.**

| Technique | 60% 1RM | | 65% 1RM | | 70% 1RM | |
|---|---|---|---|---|---|---|
| | Diff (%) | $p$ | Diff (%) | $p$ | Diff (%) | $p$ |
| | | | HBBS | | | |
| LES | −34.8 ± 7.7* | <0.001 | −33.1 ± 4.5* | <0.001 | −32.7 ± 7.6* | <0.001 |
| GM | −74.5 ± 5.1* | <0.001 | −75.4 ± 3.2* | <0.001 | −73.5 ± 3.7* | <0.001 |
| BF | −61.6 ± 3.4* | <0.001 | −60.5 ± 4.7* | <0.001 | −56.8 ± 3.8* | <0.001 |
| RF | −37.3 ± 3.0* | <0.001 | −37.7 ± 5.5* | <0.001 | −35.4 ± 4.7* | <0.001 |
| VLO | −47.5 ± 3.9* | <0.001 | −48.1 ± 3.3* | <0.001 | −43.6 ± 3.6* | <0.001 |
| VMO | −45.5 ± 3.6* | <0.001 | −46.5 ± 5.0* | <0.001 | −42.9 ± 3.2* | <0.001 |
| | | | LBBS | | | |
| LES | −33.5 ± 8.0* | <0.001 | −33.1 ± 7.8* | <0.001 | −31.4 ± 6.9* | <0.001 |
| GM | −71.6 ± 4.8* | <0.001 | −70.7 ± 5.3* | <0.001 | −69.3 ± 5.1* | <0.001 |
| BF | −64.1 ± 4.2* | <0.001 | −62.8 ± 3.3* | <0.001 | −60.0 ± 3.1* | <0.001 |
| RF | −37.8 ± 4.8* | <0.001 | −38.1 ± 6.1* | <0.001 | −34.8 ± 5.1* | <0.001 |
| VLO | −38.6 ± 5.9* | <0.001 | −41.4 ± 4.0* | <0.001 | −35.1 ± 8.1* | <0.001 |
| VMO | −38.5 ± 3.7* | <0.001 | −38.8 ± 3.6* | <0.001 | −36.5 ± 3.9* | <0.001 |

**Notes.**

HBBS, high-bar back squat; LBBS, low-bar back squat; 1RM, one repetition maximum; SD, standard deviation; Diff, difference; LES, lumbar erector spinae; GM, gluteus maximus; BF, biceps femoris; RF, rectus femoris; VLO, vastus lateralis.
*significant differences ($p \leq 0.05$) between the eccentric and concentric contractions.

60% 1RM than 70% 1RM for all muscles (HBBS and LBBS; eccentric and concentric) ($p <$ 0.05). In addition, EMG results were significantly lower during eccentric contraction than concentric contraction for all muscle groups ($p < 0.001$) (Table 3). However for the side factor, analysis demonstrated no significant differences in the bioelectrical activity values between the left and right lower extremities ($p > 0.05$).

## DISCUSSION

To the authors knowledge this study is first to investigate muscle activity differences between the HBBS and LBBS within homogeneous group during the single measuring session without repositioning of EMG probes. This methodology was chosen since changes in orientation of surface electrodes and muscle fibers are known to affect the EMG amplitude (*Paoli, Marcolin & Petrone, 2009*). In that, our approach differs from previous studies where the comparisons between HBBS and LBBS were made among different groups of lifters (*Wretenberg, Feng & Arborelius, 1996*) or it did not even include EMG activity (*Swinton et al., 2012*; *Glassbrook et al., 2017*). Another important novelty is that the EMG signal normalization was based on FBWS tests.

This study provides also notable comparisons between different barbell loads (60% and 65%, 65% and 70% as well as 60% and 70% of prior tested 1RM). Given that squat involves a symmetrical movement pattern and there were no significant differences in measured angles and bioelectrical activity between left and right lower extremities, the analysis was run only on the right side.

The main findings of this research are that the EMG bioelectrical activity during eccentric phase of squat motion for all selected muscles were significantly higher during LBBS than in HBBS (60% 1RM and 65% 1RM). During 70% 1RM squat test, those differences were also significant, except for RF and VLO. During the concentric phase, a significantly greater muscle activity was observed during LBBS for LES, GM and BF for all tested loads. These findings indicate that posterior muscles of lower extremities - hip extensors, were considerably more activated during LBBS compared to HBBS. For knee extensors, such differences were negligible and during 60% 1RM insignificant. It should be noticed that for the eccentric phase of the squat, the GM and LES muscles activated the most (both over 200% SRV during LBBS with 70% 1RM). The biggest differences between muscles activity for HBBS and LBBS eccentric phase are also demonstrated by the same muscles (GM and LES). This is due to the lower position of the bar in LBBS, which imposes higher anterior PT (Fig. 2) and a more forward trunk position, together with the wider foot stance (*McCaw & Melrose, 1999*; *Paoli, Marcolin & Petrone, 2009*).

All examined muscles were more activated during concentric phase of the SC which perfectly corresponds with previous works (*Selseth et al., 2000*; *Ebben & Jensen, 2002*; *Gullett et al., 2009*). Mean EMG activity recorded in that phase for each muscle clearly exceeds 200% SRV (for each barbell load level). Similar to the eccentric phase, the most activated muscle during concentric phase was GM (almost 700% SRV during LBBS with 70% 1RM). GM muscle activates closely twice as much as the other recorded muscle, with the exception of BF which is also very well engaged (446% SRV). What is even more notable is that GM muscle during concentric phase is activated approximately three times more than during eccentric phase for the LBBS, and almost four times more for the HBBS. This may come from more upright position of the torso in HBBS than in LBBS, which was suggested as one of the reasons for hamstring active insufficiency (due to shortened hamstring muscle length in that position), and for the presence of compensation strategy observable by greater GM activity (*Glassbrook et al., 2017*).

Unlike the eccentric phase, the biggest differences between muscle activity for HBBS and LBBS concentric phases were observed at the BF level. This proves another crucial factor in deciding which type of squat is more desirable at the moment. The reason for that may be explained by the BF stretch-shortening SC, because the wider foot stance, more forward torso position and bigger anterior PT in LBBS stretches BF more (*Gordon, Huxley & Julian, 1966*; *Escamilla et al., 2001*).

The additional goal of this study was to verify the influence of load level on the EMG activity for HBBS and LBBS. Although the load level effect on EMG activity is well described in previous studies (*McCaw & Friday, 1994*; *McCaw & Melrose, 1999*), the difference characteristics between HBBS and LBBS was still not known.

It is interesting that in the concentric phase significant differences between the loads are generally not observed between just 5% 1RM change in load level for LBBS, while for HBBS they are noticeable only between 60% 1RM and 65% 1RM. As we observed much bigger activity values for all muscles for concentric phase, this observation may indicate that for the more significant progress in LBBS training it is advisable to progress the load with at least 10% 1RM (significant differences observed for both HBBS and LBBS). This

observation only applies to the concentric phase of the SC where muscle activity level is the highest. In the eccentric phase significant differences emerged between all analyzed loads with the exception of 60% to 65% 1RM comparison for LES, VLO and VMO. Load progress affects most effectively GM and BF during eccentric phase of HBBS while the least significant influence can be observed for VLO and VMO during LBBS (concentric).

This study clearly pointed significant differences in EMG muscle activity between HBBS and LBBS. It appears that crucial differences occur for the hip extensors (GM, BF) and LES muscles. The significant differences in the EMG activity between superficially similar HBBS and LBBS is even more interesting when these results are compared with the work of *Gullett et al. (2009)*, who found no differences between totally different bar position squat techniques - front and back squat (for six muscles, unfortunately without GM). Other squat variations with variable resistance, like elastic bands or chains attached to the bar, did not provide sufficient difference in muscle activation (*Ebben & Jensen, 2002*; *Gullett et al., 2009*; *Saeterbakken, Andersen & vanden Tillaar, 2016*). For the knee extensors muscles, the difference between HBBS and LBBS is not that obvious. The differences are rather negligible which corresponds to some previous works (*McCaw & Melrose, 1999*; *Clark, Lambert & Hunter, 2012*). *McCaw & Melrose (1999)* also analyzed the activity of RF, VLO and VMO. They focused on the effect of different stance width during the parallel LBBS, but they found no change in quadriceps EMG activity either (*McCaw & Melrose, 1999*; *Clark, Lambert & Hunter, 2012*; *Clark, Lambert & Hunter, 2016*) indicated the same muscle length, as the cause of no significant differences in quadriceps EMG activity, between the various types of width stance. Because of very small differences in KFE, this argumentation may also explain minor differences for RF, VLO and VMO between HBBS and LBBS.

The potential limitation of this study was that most of the athletes were preparing for the National Academic Championships, so in order not to interfere with individual training preparations (*Issurin, 2010*), the load level was limited to 70% of 1RM. Such load level also let the athletes perform their squats with the optimum and repeatable technique, which was designed to make the comparisons between both techniques more reliable. In future studies it is worth to use similar procedure and analyse differences between HBBS and LBBS during 90% of 1RM to 100% of 1RM performances.

## CONCLUSIONS

This study is the first to compare HBBS and LBBS on the homogeneous group of experienced powerlifters. Our results confirmed the significant differences in posterior muscle chain activation between both squat techniques. LES, GM, BF, RF, VLO and VMO activity during eccentric phase of squat motion were significantly higher during LBBS than HBBS. For the knee extensors muscles, the difference between HBBS and LBBS are rather negligible. GM and BF muscles are the most crucial from tested muscles for both HBBS and LBBS but it is the LBBS which engages the muscles at the highest level. The outcomes may be useful in designing specific training programs and in optimizing performance. Our findings expand the actual knowledge providing quantitative muscular activation data. The lower bar position and the wider foot stance may remarkably influence the final result of athletes competing in powerlifting competition.

## ACKNOWLEDGEMENTS

The authors would like to thank all participating athletes. The authors declare that they have no conflict of interests.

### Funding

The authors received no funding for this work.

### Competing Interests

The authors declare there are no competing interests.

### Author Contributions

- Michal Murawa conceived and designed the experiments, performed the experiments, analyzed the data, prepared figures and/or tables, authored or reviewed drafts of the paper, and approved the final draft.
- Anna Fryzowicz conceived and designed the experiments, performed the experiments, analyzed the data, authored or reviewed drafts of the paper, and approved the final draft.
- Jaroslaw Kabacinski performed the experiments, analyzed the data, prepared figures and/or tables, authored or reviewed drafts of the paper, and approved the final draft.
- Jakub Jurga conceived and designed the experiments, performed the experiments, analyzed the data, prepared figures and/or tables, and approved the final draft.
- Joanna Gorwa performed the experiments, analyzed the data, authored or reviewed drafts of the paper, and approved the final draft.
- Manuela Galli analyzed the data, authored or reviewed drafts of the paper, and approved the final draft.
- Matteo Zago analyzed the data, authored or reviewed drafts of the paper, and approved the final draft.

### Human Ethics

The following information was supplied relating to ethical approvals (i.e., approving body and any reference numbers):

The Bioethical Committee of the Poznan University of Medical Sciences (number 546/16) granted Ethical approval to carry out this study.

### Data Availability

The raw data are available as a Supplementary File.

### Supplemental Information

Supplemental information for this article can be found online at http://dx.doi.org/10.7717/peerj.9256#supplemental-information.

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
