# Peer review of "Muscle activation varies between high-bar and low-bar back squat"

_PeerJ, doi:10.7717/peerj.9256_

## Round 0.1 · original submission · Major Revisions

The two reviewers and I see considerable merit in your study that you have described in this manuscript but also have highlighted a number of areas that need to be improved before it can be considered for publication in PeerJ.

Reviewer 1 ·

Basic reporting

The paper is generally well written, however, there are times where the English language can be improved, via sentence re-wording. Specifically, lines 63-64, line 94 (I think you mean ‘performed’ not ‘designed’), line 186, and lines 209-211.

Please check line 65, and the associated reference. I believe that the LBBS is characterised by a decreased hip moment.

Line 211-214. I am not sure your statement here is completely correct. Glassbrook et al., (2019) did compare the HBBS and LBBS within a control group.

Lines 252-255. This is a very short paragraph with only two sentences. I suggest combining with another paragraph.

Although Rippetoe is a well-known name in strength training, is it possible to replace the reference on line 251 with an academic article rather than a book reference?

Experimental design

Please provide more detail regarding your subjects. You describe them as ‘experienced powerlifters’, and also ‘experienced in resistance training and performing squats’. Were they competitive powerlifters, or specifically training for powerlifting, or just general strength trained individuals? If competitive powerlifters, what level powerlifters were they, and are the 1RM numbers provided in line 86 from competition or training?

Line 96-97. Please provide a justification of why these percentages were chosen. I’m assuming it is so that both the HBBS and LBBS can be performed in the sessions without major fatigue, however, Glassbrook et al., (2019) showed it is possible to perform 1RM for both squat variations within the same testing session.

Line 108-110. Please update figure 1 to include a representation of all 19 markers, for example: doi: 10.1080/14763140903229476 or at least provide the anatomical locations of the markers in the model. At present figure 1 does not leave me confident that your methods are capable of measuring anterior pelvic tilt. It is not clear what pelvic locations are used to measure pelvic tilt. It is only once I search into the provided references, that I find other required markers for pelvic tilt measurement represented.

Was footwear standardised at all?

Validity of the findings

No Comment

Reviewer 2 ·

Basic reporting

Figures 1 and 2 clearly illustrate the set up of the equipment and the differences in technique, which are very important to the reader. These are a great addition to the manuscript.
Line 27: Delete reference to body weight squats in the abstract
Line 44: This sentence should start, ‘The squat…’
Lines 44-48: Please add why the squat is so commonly used, based on the strong association between 1RM performance and performance in athletic tasks and the fact that numerous researchers have reported increases in athletic performance with increases in relative squat strength.
Line 59: Delete the s off the end of ‘trainings’.
Line 62: Do you mean the lower part of the upper trapezius? Based on the trapezoid shape of the muscle the lower trapezius would be halfway down the individual’s spine.
Lines 68-70: Is the HBBS chosen due to the increased forward lean of the trunk in the LBBS?
Line 70: Please re-word this sentence, as studies are inanimate and therefore cannot analyse anything.

Experimental design

Why were loads associated with hypertrophy used, when the subjects were powerlifters and the maximal strength in this populations is explained as a key performance indicator within the introduction?
Why was EMG data not normalized to permit more effective comparisons between studies?
Effect sizes should be included within the statistical analyses to highlight if any observed differences were meaningful.
Lines 96-97: A strong and clear rationale for the use of these hypertrophy based loads is essential, especially in power lifters and when maximal performance in a back squat is associated with performance in athletic tasks.
Line 97-98: Please briefly describe the protocol.
Line 120: Figure 2 does not illustrate this, there are clear differences in depth and dorsi-flexion, which are likely to affect the resultant EMG due to differences in the length tension relationship.
Line 146: It is nice to see the 95%CI included, but the results should also be interpreted based on this, with associated thresholds highlighted here, please see: Koo and Li (2016). A Guideline of Selecting and Reporting Intraclass Correlation Coefficients for Reliability Research. Journal of chiropractic medicine 15(2): 155-163.
Lines 152: It would be useful to include effect sizes.
Results: Please make any amendments based on the interpretation of the ICC’s based on the lower bound 95%CI and the addition of the effect sizes and appropriate criteria for interpretation.

Validity of the findings

Why was EMG data not normalized to permit more effective comparisons between studies?
Line 146: It is nice to see the 95%CI included, but the results should also be interpreted based on this, with associated thresholds highlighted here, please see: Koo and Li (2016). A Guideline of Selecting and Reporting Intraclass Correlation Coefficients for Reliability Research. Journal of chiropractic medicine 15(2): 155-163.
Lines 152: It would be useful to include effect sizes.
Results: Please make any amendments based on the interpretation of the ICC’s based on the lower bound 95%CI and the addition of the effect sizes and appropriate criteria for interpretation.

Additional comments

This is an interesting and generally well written study, which will be of interest to practitioners, however, there are a few areas which require attention / justification:
Why were loads associated with hypertrophy used, when the subjects were powerlifters and the maximal strength in this populations is explained as a key performance indicator within the introduction?
Why was EMG data not normalized to permit more effective comparisons between studies?
Effect sizes should be included within the statistical analyses to highlight if any observed differences were meaningful.
In the Discussion, the differences in posture and therefore muscle lengths should be explored in relation to potential differences in EMG.
Figures 1 and 2 clearly illustrate the set up of the equipment and the differences in technique, which are very important to the reader. These are a great addition to the manuscript.

---

## Round 0.2 · Minor Revisions

Thanks for your hard work in addressing the vast majority of the two reviewers concerns on the initial version of your manuscript. If you can please focus on attending to the final two comments from reviewer one that would be most appreciated.

Reviewer 1 ·

Basic reporting

"Comment 3. Line 211-214. I am not sure your statement here is completely correct. Glassbrook et al., (2019) did compare the HBBS and LBBS within a control group.
Answer 3. Indeed, Glassbrook et al., did compare the HBBS and LBBS within a control, inexperienced group but they only presented kinematic and kinetic data. The aim of our study was to compare muscle activities between HBBS and LBBS and Glassbrook et al., didn`t include EMG analysis in their research."

The sentence in the paper still doesn't reflect this. Glassbrook et al., and Wretenberg references appear before you even mention EMG. As it stands only Swinton is implied to not include EMG.

This sentence needs to be re-worded to clearly state your meaning/intention.

Experimental design

"Comment 7. Line 96-97. Please provide a justification of why these percentages were chosen. I’m assuming it is so that both the HBBS and LBBS can be performed in the sessions without major fatigue, however, Glassbrook et al., (2019) showed it is possible to perform 1RM for both squat variations within the same testing session.
Answer 7. These percentages were chosen due to the fact that some of the athletes were preparing for the National Academic Championships, so the load level was limited to 70% of 1RM in order not to interfere with individual training preparations (Issurin VB. 2010. New horizons for the methodology and physiology of training periodization. Sports Medicine 40:189–206 DOI: 10.2165/11319770-000000000-00000.) The load level limited to 70% of 1RM also let the athletes to perform their squats with the optimum and repeatable technique, which was designed to make the comparisons between both techniques more reliable.
Justification was added to the manuscript"

I can see that you have put this into a limiations paragraph, and have responded to a similiar comment from reviewer no.2. However, please add a similar (but shorter) statement in the methods section, which outlines why these percentages were chosen. A reader should be provided with this information earlier in the paper than the last paragraph of the discussion.

Validity of the findings

N/A

Additional comments

The authors have addressed the majority of my comments to a satisfactory level.
After addressing my remaining two comments, I am happy to recommend this study for publication.

Reviewer 2 ·

Basic reporting

No comment, all previous comments have been addressed

Experimental design

No comment, all previous comments have been addressed

Validity of the findings

No comment, all previous comments have been addressed

Additional comments

No comment, all previous comments have been addressed

---

## Round 0.3 · accepted · Accept

I think the authors for attending to all of the reviewers comments. I would be happy to recommend this paper being recommended for publication in PeerJ.

Reviewer 1 ·

Basic reporting

N/A

Experimental design

N/A

Validity of the findings

N/A

Additional comments

Authors have addressed my remaining comments.